# Transcriptional Regulation of the Multiple Resistance Mechanisms in *Salmonella*—A Review

**DOI:** 10.3390/pathogens10070801

**Published:** 2021-06-24

**Authors:** Michał Wójcicki, Olga Świder, Kamila J. Daniluk, Paulina Średnicka, Monika Akimowicz, Marek Ł. Roszko, Barbara Sokołowska, Edyta Juszczuk-Kubiak

**Affiliations:** 1Laboratory of Biotechnology and Molecular Engineering, Department of Microbiology, Prof. Wacław Dąbrowski Institute of Agricultural and Food Biotechnology—State Research Institute, Rakowiecka 36 Street, 02-532 Warsaw, Poland; michal.wojcicki@ibprs.pl (M.W.); paulina.grzelak@ibprs.pl (P.Ś.); monika.akimowicz@ibprs.pl (M.A.); 2Department of Food Safety and Chemical Analysis, Prof. Wacław Dąbrowski Institute of Agricultural and Food Biotechnology—State Research Institute, Rakowiecka 36 Street, 02-532 Warsaw, Poland; olga.swider@ibprs.pl (O.Ś.); marek.roszko@ibprs.pl (M.Ł.R.); 3Department of Microbiology, Prof. Wacław Dąbrowski Institute of Agricultural and Food Biotechnology—State Research Institute, Rakowiecka 36 Street, 02-532 Warsaw, Poland; kamila.daniluk@ibprs.pl (K.J.D.); barbara.sokolowska@ibprs.pl (B.S.)

**Keywords:** *Salmonella*, transcription regulators, antibiotic resistance

## Abstract

The widespread use of antibiotics, especially those with a broad spectrum of activity, has resulted in the development of multidrug resistance in many strains of bacteria, including *Salmonella*. *Salmonella* is among the most prevalent causes of intoxication due to the consumption of contaminated food and water. Salmonellosis caused by this pathogen is pharmacologically treated using antibiotics such as fluoroquinolones, ceftriaxone, and azithromycin. This foodborne pathogen developed several molecular mechanisms of resistance both on the level of global and local transcription modulators. The increasing rate of antibiotic resistance in *Salmonella* poses a significant global concern, and an improved understanding of the multidrug resistance mechanisms in *Salmonella* is essential for choosing the suitable antibiotic for the treatment of infections. In this review, we summarized the current knowledge of molecular mechanisms that control gene expression related to antibiotic resistance of *Salmonella* strains. We characterized regulators acting as transcription activators and repressors, as well as two-component signal transduction systems. We also discuss the background of the molecular mechanisms of the resistance to metals, regulators of multidrug resistance to antibiotics, global regulators of the LysR family, as well as regulators of histone-like proteins.

## 1. Introduction

Antibiotic resistance of *Salmonella* strains is among the key public health problems worldwide, both in industrialized and poorly developed countries. *Salmonella*, a Gram-negative, facultatively anaerobic bacilli belonging to the Enterobacteriaceae family, is a major foodborne pathogen that causes widespread contamination and infection worldwide [1,2,3,4]. Out of 93.8 million infections (recorded yearly) caused by consumption of contaminated food, 155 thousand cases result in the death of the patient [1,2,5].

The genus *Salmonella* consists of two species, *S. enterica*, which comprises six subspecies, and *S. bongori* (Figure 1). So far, over 2600 different serotypes of *Salmonella enterica* have been described [5,6]. The main reservoir of *Salmonella enterica* subsp. *enterica* are animals such as poultry, pigs, and cattle [7,8]. In humans, *S*. Typhi is responsible for systematic infections and typhoid fever, whereas paratyphoid is caused by the *S. enterica* of the serotypes Paratyphi A, Paratyphi B, or Paratyphi C [1,5]. The other serovars, such as *S*. Enteritidis, *S*. Typhimurium, and *S*. Agona, are associated with salmonellosis [3,6,7,8]. Typhoid fever serovars are responsible for invasive diseases in humans, while NTS (non-typhoidal *Salmonella*) serovars cause diseases in animals [3,9]. The *Salmonella* Choleraesuis serotype, originally pathogenic in animals, rarely causes systemic infections in humans [10]. The pathogenic nature of *Salmonella enterica* serovars may be specific to many hosts (e.g., *S*. Typhimurium) or limited to one host only (e.g., *S*. Typhi), and the course of infection caused by the same serovar varies and depends on the host [8,11]. For example, *S*. Typhimurium infects a wide array of hosts, both humans, and animals as well as various cell types of the host, including macrophages and non-phagocytic cells such as intestinal epithelium [8,12].

Antibiotic resistance has rapidly evolved in the last few decades to become now one of the greatest public health threats of the 21st century. The widespread use of antibiotics, especially those with a broad spectrum of activity, contributes to the development of specialist drug defense strategies by pathogens [12,13]. The mechanisms of antibiotic resistance are then disseminated in the environment by horizontal gene transfer (HGT) between bacteria and, for example, by lysogenic phages [13,14]. The primary and common mechanism of bacterial resistance to antimicrobial agents is multidrug efflux systems [15,16]. The efflux systems can transport a wide variety of structurally diverse antimicrobial agents and some metabolites out of the bacterial cell. Multidrug efflux pumps are fundamental to the physiology of different Gram-negative bacterial species and are required for virulence and biofilm formation [17,18,19]. *Salmonella* has also developed a range of molecular mechanisms of drug resistance, both at the level of global and local transcriptional regulators such as membrane sensors and/or cytosolic sensors [8,20]. In *Salmonella,* many signaling pathways regulating the expression of antibiotic resistance genes have been characterized, such as the TetR protein family (repressors that regulate tetracycline resistance genes) and AraC/XylS protein family (transcription activators/repressors), as well as two-component signal transduction systems [17,21,22,23,24]. Moreover, transcriptional regulators of mercury resistance (MerR family), regulators of antibiotic multidrug resistance (MarR family), global regulators of the LysR family (i.e., LeuO), as well as regulators of histone-like proteins (H-NS) involved in the repression of genes acquired horizontally in response to environmental factors have been also investigated [8,25,26].

In this review, we aim to interpret current knowledge about molecular mechanisms of antibiotic resistance in pathogenic *Salmonella* strains. To survive, *Salmonella* has developed complex and creative strategies to avoid antibiotic attacks. A complete investigating of mechanisms is important to design novel strategies to combat drug resistance. Future efforts will be aimed at determining the molecular networks of highly coordinated gene expression of this pathogen.

## 2. *Salmonella* Pathogenesis

The main source of *Salmonella* infection in humans is foodborne contamination [5,27]. In the first stage of infection, *Salmonella* activates the acid tolerance response (ATR) to survive in the acidic environment of the gastrointestinal tract (GT) [5,27,28] and then cross the intestinal mucosa barrier to adhere to the intestinal epithelial cells (Figure 2). During the colonization of the GT, *Salmonella* uses flagella and chemotaxis to enterocyte adhesion [29]. In cell invasion, protein SipA is one of the first proteins responsible for the induction of rearrangement of the epithelial host cell cytoskeleton. This protein causes a loss of the tightness of the host cell membrane, which facilitates efficient bacterial internalization [5]. Moreover, pathogen absorption by enterocytes leads to the formation of intracellular vacuolar compartments, known as SCVs (*Salmonella*-containing vacuoles), which are encoded on two pathogenicity islands (SPIs, *Salmonella* pathogenicity islands), SPI-1 and SPI-2, respectively [27]. *Salmonella* transfers over 30 effector proteins coded by SPI-2 or by virulence factors through the SCV membrane, thus enabling the pathogen to replicate, colonize, and establish an infection inside the host [30]. In the host, pathogen invasion induces the gene expression of the pro-inflammatory cytokine proteins (i.e., of IL-1, IL-8, IL-10, and IL-18), which results in an inflammatory response. It leads to the migration of polymorphonuclear leukocytes (PMN) to the small intestine and induction of antimicrobial substance synthesis, to which *Salmonella* is much more resistant than most of the physiological gut microflora [7,29]. Inflammation caused by pro-inflammatory cytokines leads to diarrhea, ulcers, and the destruction of mucosa cells, resulting in local inflammation of the intestines [31]. During an infection, *Salmonella* uses various virulence factors, such as lipopolysaccharide (LPS) and pili (fimbria), synthesis of enterotoxins, and cytotoxins, as well as the presence of numerous SPIs [29,32]. LPS stimulating the release of pro-inflammatory cytokines specific for *Salmonella* is mainly responsible for clinical symptoms of infection [33]. The external part of the LPS plays an important role in the host colonization; length, structure, composition, and roughness of the surface of O side chains in LPS determine the pathogen virulence [28,29,33]. In humans, *Salmonella* uses type I fimbria, including long pole fimbria (Lpf) and thin aggregation fimbria (Tafi), enabling adhesion to enterocytes, while type IV pili are used by *S*. Typhi to invade host cells [29].

*Salmonella* induces invasion of M cells (by adhering to the apical side of M cells), or enterocytes using the type III secretion system encoded within SPI-1 [29,34,35]. This needle-like complex enables the introduction of bacterial proteins (effectors) into the cytoplasm of epithelial cells. Effectors cause actin reorganization, leading to bacteria absorption and inflammatory response induction [28,29,34].

SPIs are chromosomal regions transferring virulence genes and play an important role in *Salmonella* pathogenicity [5,36]. The functions of SPIs in *Salmonella* are presented in Table 1.

SPI-1 and SPI-2 represent two of the SPIs crucial for *Salmonella* pathogenicity [44,48,49,50]. SPI-1 is required to invade intestinal epithelium cells, induce an inflammatory reaction, and violate the epithelium barrier in the host [33,38]. SPI-1 includes genes involved in effector protein transcription, which are pivotal in the rearrangement of the host actin cytoskeleton [12]. Expression of SPI-1 genes is induced under the influence of environmental stimuli present in the gastrointestinal tract, i.e., low oxygen levels, high osmolarity, and neutral pH [33]. SPI-2 enables survival and growth of the pathogen in macrophages, leading to a general systemic condition [37,39]. SPI-2 is also responsible for SCV location and inhibits the fusion of lysosomes and SCV [40]. SPI-2 contains genes that code effector, chaperones, and translocon proteins [12] inducted under the influence of environmental stimuli present in macrophages; low levels of inorganic phosphate (Pi) and Mg^2+^ ions, as well as lightly acidic pH [33]. Moreover, SPI-2 genes codes a type III secretion system (similar to SPI-1) and transfers over 30 unique protein effectors to host cells during infection [37]. These proteins play an important role in protecting the pathogen against the immune response of the host by modification of intracellular vacuoles [51]. The level of the SPI-2 gene expression is controlled by several TCSs (which are discussed in Section 3.3.) [51]. Furthermore, in *S.* Typhimurium, SPI-3, SPI-6, SPI-11, SPI-12, SPI-13, and SPI-16 are responsible for pathogen survivability inside multinuclear cells and induction of a general systemic infection [38]. SPI-4 codes the *siiA-F* genes involved in intestine infection caused by *Salmonella* and the non-fibrous adhesion protein SiiE [37]. SPI-5 (similar to SPI-1) is associated with membrane wrinkling and activation of pro-inflammatory response [12,38], whereas in *S*. Typhi, SPI-7 transfers the Vi capsular antigen biosynthesis gene, which is a crucial virulence factor of typhoid fever [46]. For SPI-14 encoded in *Salmonella*, the virulence-regulating protein LoiA, belonging to the LysR family (LTTR, LysR-type transcriptional regulator), plays an essential role in pathogen adhesion to the intestinal epithelium [37,40]. Furthermore, SPI-14 is responsible for the regulation of SPI-1 genes [38]. Xian et al. [47] reported that a type VI secretion system (T6SS) encoded in SPI-19, presented in *Salmonella enterica* subsp. *enterica* serovar Pullorum (*S*. Pullorum) is a virulence factor necessary for pathogen survival in macrophages and initial colonization of chickens. The pathogenicity of *Salmonella* is also determined by plasmids carrying genes encoding virulence factors [52,53]. In *S*. Typhimurium, *S*. Dublin, and *S*. Enteritidis, these factors are responsible for the general systemic spread of infection in lymph nodes of the mesentery, spleen, and liver [29]. Moreover, the pathogenicity of *Salmonella* strains is associated with the ability to synthesize two toxins, such as enterotoxin and cytotoxin, secreted into the host cells [50]. Enterotoxin induces liquid accumulation in the ileum and exhibits cytotoxic properties [29,54]. Cytotoxin inhibits protein synthesis and is responsible for damage to the intestinal mucosa surface, leading to inflammatory diarrhea [5,12].

The pathogenicity of *Salmonella* is regulated by short- and long-chain fatty acids formed both by the host and gut microflora [55,56,57]. It was shown that butyric and propionic acids present at high concentrations in the intestine, as well as oleic acid present in bile, can inhibit the virulence of *Salmonella*. Bosire et al. [57] proved that cis-2-unsaturated fatty acids (c2-HDA, cis-2-hexadecaenoic acid) were used as diffusible signal factors (DSF) are strong virulence inhibitors. Aromatic compounds (e.g., aromatic acids, phenolic compounds) also influence the formation of flagella and inhibition of motility [58,59,60]. In the case of aromatic acids, to which the cytoplasmic membrane is permeable (e.g., benzoic acid), the influence on mobility is related to the interruptions of the proton driving force during the passage of such molecules through the cytoplasmic membrane. Some phenols (e.g., curcumin) bind to flagella monomers and induce their separation and loss of mobility in bacteria [59,60]. Activation of MarA, SoxS, RamA, or Rob is a common response to many aromatic compounds [8,58].

In *S.* Typhimurium (but also for other serovars of the *S*. *enterica*), flagella formation is controlled by induction of *flhDC* transcription and translation [58]. The heteromeric transcription factor FlhD_4_C_2_ expressed from the *flhDC* operon (class I genes) is the main regulator of flagella genes (class II genes) expression, such as FliA. It is known that FliA initiates the expression of class III genes: late flagella and chemotaxis genes [61,62]. In addition to regulating transcription, several feedback mechanisms provide checkpoints during flagella folding (e.g., regulation of bonding or stability of FliA and FlhD_4_C_2_). In *S*. Typhimurium, *flhDC* expression is regulated by several factors such as cAMP (CRP) receptor protein, Fur, proteins binding Fis nucleoid, histone-like protein structuring the (H-NS) nucleoids, and SlyA [63]. HilD is a crucial regulator of the SPI-1 that activates *flhDC* transcription, whereas several regulators, including RtsB, LhrA, OmpR, SsrB, and RcsB, attenuate *flhDC* expression [58].

## 3. Molecular Systems of Multiple Resistance in *Salmonella*

The development of multidrug-resistant (MDR) in *Salmonella* has a significant influence on antibiotic therapy against this pathogen [1,64,65]. Fluorinated quinolones (fluoroquinolones, FQ) are commonly used in salmonellosis therapy [66,67]. The mechanism of quinolone activity is associated with inhibition of DNA synthesis by blocking topoisomerases II, DNA gyrase, and topoisomerase IV [66,67,68,69,70,71]. Another approach in salmonellosis therapy is treated with ceftriaxone or azithromycin [72,73,74].

Mechanisms of *Salmonella* resistance are controlled via the expression regulation of the genes encoding proteins related to drug transport [73,75,76,77]. The first line of pathogen defense involves membrane resistance based on limited influx through porins of the external membrane and on increased export by membrane transporters [78,79]. The second line of pathogen defense includes the change of antibiotic targets caused by mutations of the gyrase and topoisomerase IV genes as well as enzymatic drug decomposition affected by bacterial ß-lactamases [8,80].

### 3.1. The TetR Family

The TetR proteins control the expression of the genes involved in multidrug resistance and pathogenicity of both Gram-negative and Gram-positive bacteria [17,81]. The main function of these transcription repressors is to control processes important in the acquisition of bacteria resistance to antimicrobial agents such as drug export regulation [82,83]. TetR proteins are implicated in different catabolic pathways, the division of bacterial cells, and osmotic stress response [17]. Twelve genes coding TetR regulators have been identified in the *S*. Typhimurium and *S*. Typhi, and 14 in the *S*. Choleraesuis [82,84]. Moreover, Colclough et al. [82] reported the presence of nine TetR-encoding genes (i.e., *acrR, envR, nemR, slmA, ramR, rutR, ycfQ, yjdC*, and *U1*) in various *Salmonella* strains. RamR, AcrR, and EnvR belonging to the TetR family play in *Salmonella* a pivotal role in the regulation of multidrug exporting systems [20,82]. The RamR binding site is located in its promoter, where it overlaps with the divergent *locus* located below and coding another regulator, RamA (an AraC family regulator, discussed in Section 3.2) [8,17,85,86]. In *S*. Typhimurium, susceptible to antibiotics, inactivation of the *ramR* gene results in acquired drug resistance [8]. Moreover, it has been reported that RamR downregulates the expression of both *ramA* and *ramR* genes [82]. The RamR-induced repression mechanism involves the binding of dimeric RamR to a 28 bp target site that includes the conserved ramA promoter region [8,87]. It has been reported that mutations in the *ramR* gene result in repressive relaxation of *ramA*. Fàbrega et al. [68] showed that mutations of the *ramR* have a smaller impact on the relaxation of RamA repression compared to changes to the binding site in the region of the *ramA* promoter. In the MDR strains of *S*. Typhimurium and *Salmonella enterica* subsp. *enterica* serovar Kentucky (*S*. Kentucky), several mutations associated with RamR repression have been identified [8,68].

Yamasaki et al. [88] reported that bile salts inhibit the binding of RamR to the *ramA* promoter and activate the expression of the *ramA* gene, increasing the *acrAB* and *tolC* expression. Baucheron et al. [86] proved that the *ramR* and *ramA* genes are co-activated by bile because of the divergently overlapping promoters. RamA is the main activator of *acrAB* and *tolC* transcription, and bile induces an over two-fold increase in *acrB* and *tolC* transcript levels. Moreover, the induction of the expression of genes involved in export by bile salts depends on the *ramA* gene [86]. In the case of bacterial *ramR¯/¯* mutants, it was found that bile significantly increases *ramA* expression, suggesting that stress-induced pathways, independent of RamR, also participate in *ramA* induction [86]. Studies by Yamasaki et al. [20] showed that the binding of many drugs with RamR decreases its affinity to DNA, thus inducing *ramA* expression. The high concentration of drugs may alleviate RamA suppression via RamR saturation. It has been shown that RamA negatively influences the virulence of *S*. Typhimurium by decreasing the expression of pathogenicity island genes, depending on the genetic background of the strain [86].

In *S*. Typhimurium has been identified at least nine multidrug export pumps, one of which, AcrAB, is important in drug resistance [17,89,90]. The regulators of this pump are members of the TetR family [17,91]. The regulatory pathways of the AcrAB-TolC activity are shown in Figure 3.

The structure of the AcrAB pump includes components of the transporter protein in the inner membrane (AcrB), the auxiliary periplasmatic protein (AcrA), and the external membrane channel (TolC) [92,93]. AcrB recognizes and binds substrates in the phospholipid bilayer and transports them to the external environment via the TolC protein [82]. The cooperation between AcrB and TolC is intermediated by the periplasmatic protein AcrA. The level of the *acrAB* expression is controlled by *acrR* (MarRAB operon), including the positive MarA regulator or the SoxRS operon [94]. AcrR is an important regulator of the AcrAB belonging to the TetR family [71,82,95]. Mutations of the *acrR* gene are associated with the increase in the AcrAB expression and acquisition of drug resistance by pathogenic strains of *S*. Typhimurium [68,96]. Nikaido et al. [95] reported that induction of *acrAB* and *ramA* expression in response to indole depends on RamR.

**Figure 3 pathogens-10-00801-f003:**
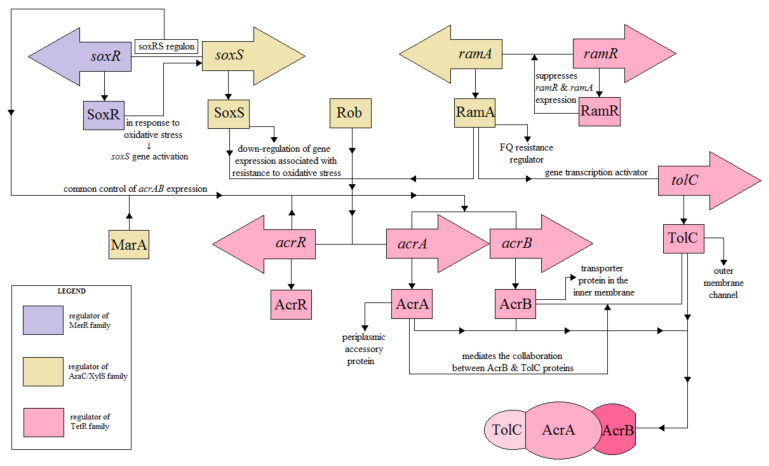
Schematic representation of the known regulatory pathways of the AcrAB-TolC efflux pump expression in *Salmonella* (adapted from on Grimsey et al. [93]).

Three homologous transcriptional activators, RamA, SoxS, and MarA, controlled by RamR, SoxR, and MarR, are involved in activating acrB and tolC expression [68]. In *S*. Enteritidis, deletion of the *soxS, marA,* and *ramA* showed an increase in the *acrB* expression, which indicated the contribution of these regulators in the resistance to FQ and MDR [97].

In *Salmonella, ramA* regulates the multidrug resistance, which is intermediated by AcrAB, regardless of the *marA* regulator [98,99]. Zhang et al. [98] demonstrated the direct contribution of marA and soxS to the regulation of AcrAB-dependent multidrug resistance in *Salmonella*. The ∆soxS mutant showed a deeper downregulation of *acrB* and lower MIC values in the study of antibiotic resistance compared to the ∆ marA mutant. In the case of MDR phenotypes, the clinical significance of mutations located in genes coding RamR, SoxR, and MarR regulators was confirmed only for RamR [68]. MarA overexpression was detected in MDR strains of *S. enterica*; however, no functional mutations in the coding region of the *marRAB* were found [68]. The *acrB* inactivation results in increased expression of the *acrAB* operon. It has been shown that ∆*acrA* or ∆*tolC* mutants caused an increase in the *acrB* expression, while ∆*acrAB* induced transcription of eight functional genes of export pumps (i.e., *acrF, acrD, mdsB, mdtB, macA, emrA, mdfA*, and *mdtK*) [98].

### 3.2. The AraC/XylS Family

AraC/XylS family (or AraC) include approximately 830 regulators identified in Gram-negative bacteria [21]. In *Salmonella*, AraC controls the expression of many virulence factors, especially those required for adhesion and colonization [21,100]. In *Salmonella*, the AraC negative regulators (ANRs) family is responsible for inhibition of the expression of the virulence genes via repression of AraC/XylS [21,101]. The AraC has been divided into three functional categories [102]. The first group includes proteins regulating genes involved in carbon metabolism (e.g., AraC). Regulators of this group usually operate as dimers for induction of target gene transcription, with the N-terminal domain responsible for dimerization and binding of the regulator ligand. The second group including MarA, SoxS, and Rob regulators, which are involved in the detection of environmental stress and recognize and bind similar sequences, known as Mar-Rob-Sox cassettes, to induce or repress transcription of their target genes [21]. The MarA and SoxS regulators contain only the C-terminal DNA binding domain and do not have a separate N-terminal domain to act as a transcription regulatory monomer for the target genes. The third group includes proteins regulating the expression of the virulence genes in response to various environmental stimuli. In contrast to the two previous groups of AraC regulators, there is small conformity between the oligomeric state of these proteins between bacteria of different species [103]. RamA is a global regulator of AraC transcription and the main regulator of *Salmonella*-specific fluoroquinolone resistance [89].

In *S*. Typhimurium, AraC-like regulators, HilC, and HilD have been involved in the control of HilA expression and other genes, including lipid A deacylase [21]. Moreover, HilD induces *hilA* expression by forming a feedback loop with the transcriptional activators, HilC and RtsA, belonging to the AraC/XylS family [58]. HilA is responsible for the activation of the expression of SPI-1 and T3SS genes coding the needle complex and secretion of the effector protein required for invasion of the host intestine [57,58]. The needle complex of the type III secretion system includes protein-like rings supporting the needle fiber, stretching to the extracellular environment, and used as the main channel for effector protein translocation [104].

Another member of the AraC/XylS involved in the regulation of AcrAB-TolC expression is RamA belonging to the resistance nodulation division (RND) efflux pump family [86,89,95,105]. In *Salmonella,* RDN plays a pivotal role in virulence and stress response [89,95]. In *S*. Typhimurium, RamA imparts resistance via activation of MdtK, a MATE (multidrug and toxic compound extrusion) family transporter, and overexpression of *acrAB* [8]. Nikaido et al. [89] showed that induction of the *acrAB* via indole is regulated by the RamA. In *S*. Typhimurium, the bile-inducted expression of the multidrug export genes such as *acrAB* and *tolC* occurs mainly as a result of transcriptional derepression of *ramA* [86]. Zheng et al. [106] reported that the constitutive expression of *ramA* is directly related to the increase in the AcrAB-TolC expression and the decrease in the OmpF porin expression, determining the phenotype of resistance to many drugs.

SoxS protein is an important regulator of resistance acquisition of *Salmonella* [84] and, together with SoxR, forms a collective soxRS regulon [25,84], which is associated with cellular defensive systems in response to oxidative stress [25]. In *S*. Typhimurium, SoxS downregulates the expression of genes involved in resistance to oxidative stress; however, it is a positive regulator of SPI-2 genes [8,25]. SoxS participates in drug resistance mechanisms by inducing the expression of AcrAB in response to SoxR activation [25,102], which is induced by methyl viologen as a result of indirect DNA oxidation [95].

Expression of the MarA, SoxS, and RamA is controlled by MarR_2_, SoxR_2_, and RamR_2_, respectively, with each of them being responsible for a different compound [58]. For example, the MarR_2_-dependent transcriptional repression of the *marRAB* operon (discussed in Section 3.5) is moderated by MarR_2_ binding with aromatic acids or the formation of disulfide bonds between MarR monomers via copper [107]. The SoxR_2_-dependent activation of *soxS* transcription occurs via oxidation of the iron-sulfur cluster in SoxR_2_ by redox-active compounds, such as methyl viologen [58]. Similar to MarR_2_, RamR_2_ is responsible for *ramA* inhibition to bile salts exposure and other aromatic compounds, such as indole [58,88]. Contrary to other regulators, the Rob is post-translationally activated via a sequestration and dispersion mechanism in response to aromatic and fatty acids [58]. In addition to efflux outside the cell (efflux pumps), conversion to non-toxic compounds, and degradation, the sequestration mechanism is one of the mechanisms developed by pathogens to reduce the harmful activity of some compounds (e.g., free haem excess) [108]. Extensive regulatory communication exists between these systems, making it possible to create complex feedback control loops depending on chemical inductors. For example, exposure to salicylic acid activates *marA* expression, leading to more potent activation of mar-sox-rob targets [109].

Pathogens containing *mar-sox-rob* regulatory networks can precisely tailor further reactions to a wide range of chemical stressors in the environment, based on intracellular concentrations of MarA, SoxS, Rob, and RamA [58]. These regulators are motility repressors with the strongest phenotypic activity of the Sox and RamA during transcription of the flagellar regulon. Interestingly, the repressive influence of SoxS on mobility is a result of both transcriptional and post-transcriptional regulation of *flhDC* [58,110]. Flagella repression occurs via coordinated activation of MarA, SoxS, Rob, and RamA in the presence of various chemical stressors [110]. This form of repression occurs via *flhDC* repression and activation of the post-transcriptional mechanism inhibiting *flhDC* translation [58]. MarA, SoxS, Rob, and RamA present in the gastrointestinal environment are expressed at different stages of *S. enterica* infection, and the mechanism of flagella gene expression may affect the virulence of this pathogen [58,111].

### 3.3. Two-Component Signal Transduction Systems

The ability of *Salmonella* to react to external stimuli is controlled by a two-component signal transduction system (TCS) [22,23,24]. The TCS system responds to a specific environmental signal such as pH, levels of nutrients and oxygen, quorum-detecting proteins, osmotic pressure/osmolality, or the presence of antibiotics [112,113,114]. In *Salmonella*, several TCS systems associated with antibiotics resistance, such as PhoPQ, CpxAR, BaeSR, EnvZ/OpmR, and PmrAB, have been identified (Table 2). TCSs consist of sensor proteins (usually histidine kinase bound to the membrane) and a cytoplasmic response regulator (RR) [8,51,115]. The histidine kinase (HK) has two domains, a variable N-terminal entry domain and a conservative C-terminal domain interacting with the RR [51]. The N-terminal domain of HK is the most variable because it defines the specificity of the environmental ligand and includes one to several transmembrane domains [23]. The cytoplasmic response regulator (RR) contains a conserved N-terminal receptor domain and a variable C-terminal exit domain. Histidine kinase is a transmembrane protein (homodimer); thus, signal detection and transmission domains are located in all three cell compartments, such as extra-cytosol space, the membrane, and the cytoplasm [22,114].

#### 3.3.1. The PhoP-PhoQ System

The best-known TCS-inducing virulence phenotype of *Salmonella* is the PhoP-PhoQ system [116], including the sensor kinase (PhoP) initiating the phosphorylation process, which is required for activation of RR-PhoP transcription in response to environmental signals [115]. Activation of PhoPQ signaling determines the acquisition of resistance to cationic peptides as a result of weaker peptide binding and the variable hydrophobic nature of the membrane [123]. PhoPQ is activated by a range of external factors, i.e., low levels of bivalent cations (e.g., Mg^2+^), low pH, the presence of CAMP, or hyperosmotic stress [116,117,118]. Under stress conditions, pathogens modify their LPS, increasing resistance both to conventional antibiotics and antimicrobial peptides (AMP) [124]. Activation of the PhoPQ system modifies the region of the A lipid of LPS through an additional palmitoyl chain, a hydroxyl group, and a positively charged amino arabinose residue. These modifications lead to increased bacterial resistance to polymyxins and lipophilic drugs [116]. In *Salmonella*, PhoPQ is the main activator of the type III secretion system (T3SS) coded in SPI-1 [125]. PhoP negatively regulates *hilA* via direct repression of *hilA* transcription, indirect repression of both *hilD* and *rtsA* expression, and activation of small RNA (sRNA) PinT. PhoP specifically binds the *hilA* promoter to block the binding of HilD, HilC, and RtsA activators as a repression mechanism [126].

#### 3.3.2. The CpxAR System

In *Salmonella*, CpxAR includes a histidine kinase sensor (CpxA) and response regulator (CpxR) [124]. The third component of this system is a periplasmic protein (CpxP), which acting as a CpxA repressor by its binding [124,127]. CpxAR regulates the expression of the *htrA, dsbA*, and *ppiA* involved in protein folding and degradation in the periplasm [124]. Huang et al. [11] evaluated the important role of the CpxR regulators in the acquisition of resistance to aminoglycosides and β-lactam antibiotics in *S*. Typhimurium via expression regulation of genes related to MDR (i.e., *marA* and *soxS*). Moreover, the lower activity of the SoxS and MarA regulators and decrease in *acrB* expression was observed, which suggest that CpxR can decrease *acrB* expression via SoxS and MarA regulation [11]. Another study [128] confirmed that antibiotic resistance is acquired in a CpxR-dependent manner through regulation of the OmpF porin and of the AcrD efflux pump. The AcrAB efflux pump contains an outer membrane protein (TolC), cooperating with many transporters, such as AcrD and MdtABC, to remove drugs from pathogen cells [129]. The outer membrane protein STM3031 plays a key role in the acquisition of bacteria resistance to ceftriaxone [130] by decreasing membrane permeability as a result of decreased OmpD level and increased export caused by the activity of the AcrD efflux pump. Zhai et al. [119] showed that in *S*. Typhimurium, the AcrAB-TolC efflux pump and CpxR regulate transcription of colistin resistance-related genes (CRRGs), binding directly to the *phoPQ, pmrC, pmrH*, and *pmrD* promoters of *box-type* CpxR sequences, or indirectly to *pmrAB* and *mgrB*.

#### 3.3.3. The BaeSR System

The BaeSR system includes a sensor kinase (BaeS) and a response regulator BaeR [131,132]. In *S*. Typhimurium, BaeSR is responsible for the activation of two multidrug efflux pumps (MdtABC and AcrD) [8,131]. Guerrero et al. [133] showed that in *Salmonella*, *mdtA* expression is regulated directly by BaeR binding in the promoter region, and this interaction is strengthened by protein phosphorylation. Moreover, the BaeSR system positively regulates its own transcription because the *baeSR* gene is located exactly below the *mdtABCD*, forming a shared operon [8]. Both TCS systems discussed above (i.e., CpxAR and BaeSR) are susceptible to ceftriaxone. Elevated levels of STM3031, STM1530, and AcrD and decreased OmpD were correlated with the ceftriaxone resistance phenotype [129].

#### 3.3.4. The EnvZ/OmpR System

The EnvZ/OmpR system consisting of the histidine kinase sensor (EnvZ) and a response regulator (OmpR) is activated after the pathogen enters macrophages of the host as a result of changes to medium acidity [131,134]. EnvZ reacts to intracellular acidic stress [121]. EnvZ, activating the RR, regulates the expression levels of outer membrane porins (OmpC and OmpF), depending on the level of chemicals in the environment. In *S*. Typhimurium, a decreased expression of genes coding these proteins induces resistance to β-lactam antibiotics [131]. Moreover, OmpR inhibits the *cadC/BA* system, preventing the neutralization of bacterial cytoplasm [135]. As a result, OmpR and PhoP activate transcription of *ssrA* and *ssrB* genes, leading to the formation of SsrA and SsrB. During acidic stress, PhoP increases the level of SsrB. In the presence of the SsrA kinase, the phosphorylated SsrB protein drives the expression of SPI-2 genes by derepressing H-NS proteins and activating *ssaB, sseA, ssaG*, and *ssaM* transcription [121]. SsrB also induces the gene expression of the type III secretion system (T3SS) and secretion effectors such as SifA and SseJ [135,136]. At neutral pH, the SsrA kinase is present at low concentrations, and the non-phosphorylated SsrB form dominates, driving the expression of genes responsible for biofilm formation [121].

#### 3.3.5. The PmrAB System

The PmrAB system has been identified in a mutant strain of *S*. Typhimurium LT2 associated with resistance to polymyxin B (PMB) [122,137,138]. Expression of the *pmrCAB* operon results in three protein products: a phosphoethanolamine (pEtN) phosphotransferase (PmrC), a response regulator (PmrA), and a sensor kinase (PmrB) [137,138].

In *Salmonella*, PmrAB is one of the main regulators of LPS modification [122,137]. External signals such as high Fe^3+^ and Al^3+^ concentration and low pH induce autophosphorylation of the PmrB in the conserved histidine residue and transfer of the phosphoryl group to the conservative asparagine PmrA residue [139]. Active PmrA (PmrA-P) induces transcription of genes involved in lipid A modification of the LPS structure. PmrA has two main domains: the N-terminal receptor domain and the C-terminal DNA binding domain [137]. Iron ions bind directly to the periplasmic domain PmrB, which contains two copies of the ExxE motif. This motif is also required to respond to high aluminum ion concentrations, although the detailed mechanisms behind its signaling are not known [122]. Direct detection of a mild decrease in medium acidity by PmrB in *Salmonella* requires the presence of a single histidine residue and four glutamate residues in the periplasmic domain. A decrease in the pH of the medium is associated with activation of the *eptA* and *arnBCADTEF-ugd* expression, which is regulated by PmrA [122,137]. Moreover, PmrAB may also be activated indirectly by PhoPQ (the previously discussed TCS system) [122].

PmrAB modifies the bacterial LPS by adding Ara4N and pEtN to the A lipid and pEtN to the LPS core [122,137]. This masks the phosphate groups of positively charged AMP, which affects the electrostatic interaction of some cationic antimicrobial peptides with the surface of the bacterial cell [122]. PmrAB regulates the pmrH expression, the first gene in the seven-gene operon associated with the addition of Ara4N to lipid A, which is expressed early in *Salmonella* infection in response to unknown factors in vivo [140]. Ara4N is added to the 4′-phosphate of lipid A (sometimes in position 1), whereas pEtN may also be added to position 1 (and to position 4′ in a mutant unable to add Ara4N). Modification of the cell surface charge increases resistance to ions and congenital resistance factors (e.g., antimicrobial peptides) [122].

### 3.4. The MerR Family

Several various regulators of the MerR family were identified in genomes of different *Salmonella* serovars [8]. The MerR family is a group of transcription activators consisting of region-binding DNA in the N-terminal domains containing HTH motifs and C-terminal region binding the effector, specific to the identified effector [25,141,142]. MerR family is characterized by the similarity of amino acids in the first 100 amino acid residues, including the HTH motif followed by the spiral region. Transcription activation occurs through protein-dependent DNA deformation [141]. Most regulators of this family react to environmental stimuli, such as oxidative stress, heavy metals, or the presence of antibiotics [142]. The SoxR regulator is a well-known transcription activator of the MerR family (its indirect role in antibiotic resistance related to activation of the soxS gene expression was discussed in Section 3.2) [8,25].

A subgroup of the MerR family activates gene expression in response to metal ions and displays sequence similarity in the C-terminal region binding the effector as well as in the N-terminal region [142]. Heavy metals are toxic to all living organisms, with mercury being one of the most harmful elements of the group. Microorganisms have developed several mercury removal mechanisms, the most common of which is the reduction in Hg^2+^ to Hg^0^ [8]. MerR regulators control gene expression via the DNA modification mechanism [141]. The binding of metal ions at the C-terminal binding site of the inducer causes an allosteric change in the N-terminal DNA binding region of the protein leading to changes in the structure of the promoter leading to the induction of the genes expression involved in the efflux effect or detoxification system [142,143]. The MerR protein negatively regulates both its own synthesis and expression of the polycistron *mer* operon in mercury detoxication [141,142]. In a mercury-free environment, transcription of suitable genes is inhibited because their promoter region is modified by adding an apo-MerR homodimer. Binding of the mercury ion, the Hg^2+^-MerR complex by MerR causes allosteric addition to the promoter region and unblocking of the sequences at positions -35 and -10, allowing σ^70^ RNA polymerase binding and transcription initiation [141,142].

### 3.5. The MarR Family

The MarR family includes homodimeric proteins such as MexR, SarR, SlyA-Ef, AphA, and OhrR containing in their structure six α-helices and three β-strands [144]. The N- and C-terminal helices form a compact dimerization interface between the subunits via hydrophobic interactions and intermolecular hydrogen bonds [145,146]. It has been shown that mutations in the dimer interface and the C-terminal region of the MarR decreased the affinity of DNA binding and the ability to form dimers, resulting in the acquisition of drug-resistant phenotypes [145]. The SlyA protein is one of the MarR family members identified in *Salmonella* [147]. SlyA plays an important role in *Salmonella* virulence, resistance to oxidative stress, and antimicrobial proteins [144,148]. SlyA regulates gene expression in response to a variety of molecules, including antibiotics, organic compounds, disinfectants, and H_2_O_2_ [147,148]. Moreover, SlyA controls the expression of genes coding outer membrane and periplasm proteins as well as secretion proteins that are implicated in virulence and bactericidal peptides resistance (i.e., *nmpC, pagC, ugtL*, *mgtB*). In *S.* Typhimurium, RNA-seq analysis showed that under oxidative stress, the SlyA regulated both positively and negatively the expression of 83 genes involved in the replication and pathogen survival [147]. Furthermore, Okada et al. [144] reported that the SlyA protein promotes the SPI-2 by activating the SsrA/SsrB regulatory system as a result of regulating the *ssrA* expression. The inability of the *slyA¯/slyA¯ Salmonella* mutant to survive in macrophages of the host was associated with a decrease in the SPI-2 gene expression.

MarR represses the expression of the marRAB operon, which encodes MarR (MarR family) and MarA (AraC/XylS family, discussed in Section 3.2). These regulators control the *marRAB* operon, both by negative and positive regulation. The *marRAB* operon is also induced by SoxS and Rob, which are MarA homologs [97]. MarA is responsible for *Salmonella* drug resistance by the regulation of the AcrAB efflux system. Deoxycholate binding with MarR prevents its binding with DNA in *Salmonella* and thus mitigates depression of *locus marRAB*. The *marA* derepression results in overexpression of AcrAB, including repression of OmpF. O′Regan et al. [97] reported that *ramA* acts as the main regulator of *Salmonella* drug resistance, controlling both *soxS* and *rob* expression. Ballesté-Delpierre et al. [149] demonstrated that fluoroquinolone-resistant mutants of *S*. Typhi with AcrAB-TolC overexpression were assigned change in the regulatory region of the *marRAB* operon. The change included the deletion of almost the entire *marR* gene, which resulted in a lack of repressor transcription, leading to MarA overproduction. The increased production of MarA and AcrAB leads to internal accumulation of the drug by decreasing the expression of porins (OmpF and OmpC). The *marRAB* operon and *soxRS* genes are involved in the OmpF repression control [97].

### 3.6. The Histone-Like Protein Family (H-NS)

Bacteria achieve genetic variabilities, such as virulence or antibiotic resistance by horizontal gene transfer (HGT) [13,14]; however, incorrect expression of the newly acquired genes may be harmful to the bacteria. Global transcriptional regulator, H-NS, helps bacteria survive stressful conditions by suppressing the expression of acquired genes [26,150]. The H-NS proteins are involved in the suppression of genes acquired horizontally as a response to environmental factors, such as temperature, pH, and osmolarity [8,26]. Regulation mechanisms are adapted to induce the expression of acquired genes in individual niches to receive benefits from information coded in alien DNA, as is the case with pathogenesis [26].

The N-terminal domain of H-NS includes 89 amino acids and four α helices (H1-H4). Two dimerization sites are present in H-NS, which enable head-to-head and tail-to-tail binding, one near H2 and the other between H3 and H4. These interactions lead to the formation of the H-NS chain and the superhelical protein skeleton. The C-terminal of H-NS contains 47 amino acids and a small loop, which may interact with smaller grooves of AT-rich DNA, acting as a global transcription regulator [150].

In *S*. Typhimurium, H-NS regulates over 400 genes, including pathogenicity islands of *Salmonella* (SPI-1 to SPI-5) [150,151]. For example, a two-component regulatory system, SsrA/SsrB, that regulates the type III secretion system, encoded by SP-2, is repressed by the H-NS binding [152]. In the virulent stage of the pathogen, HilD positively regulates the expression of the *ssrA/ssrB* by counteracting H-NS. On the other hand, many genes acquired as a result of HGT are regulated by the two-component PhoP/PhoQ system and by H-NS in *S*. Typhimurium [150]. To identify foreign genes, the H-NS system recognizes a prokaryotic AT-hook DNA binding motif, which is preferentially inserted into DNA sequences rich in AT pairs [152]. Thus, these proteins bind the newly acquired sequence and prevent the potentially harmful effects of their uncontrolled expression [151,153]. In macrophages, *S*. *enterica* degrades H-NS allowing the expression of HGT. A decrease in H-NS levels due to H-NS proteolysis by the Lon protease leads to the derepression of alien genes, even those not bound by anti-silence DNA-binding proteins [151]. Conservation in amino acid sequences of both the Lon protease and in the H-NS itself suggests that gut microflora have a shared, discovered strategy of expression of alien genes suppressed by H-NS. The H-NS is protected before Lon when it is bound to DNA [153].

The PhoP protein, phosphorylated in low Mg^2+^ concentrations [150], promotes H-NS degradation through H-NS displacement from DNA. The activity of PhoP enables HGT transcription, even of those without a PhoP binding site in their promoter regions [154]. In *Salmonella*, *pagC* and *ugtL* are bound by H-NS proteins. PagC is an outer membrane protein involved in virulence, while UgtL is a cell membrane protein regulating the formation of monophosphorylated A lipid in lipopolysaccharide and contributes to resistance to some antimicrobial peptides [150]. The ability of H-NS proteins to self-associate and form elongated protein fibers along target sequences is of key importance in this process. H-NS self-association is achieved via N- and C-dimerization domains, where the C-terminal is responsible for DNA binding activity [151]. Nucleoprotein filament formation is thought to occur in a cooperative manner whereby H-NS DNA binding initiates a single high-affinity site formation and is extended by polymerization of local H-NS molecules along the DNA sequence [155].

Ali et al. [151] determined the crystalline structure of Hha (a protein regulating hemolysin expression) in a complex with the N-terminal dimerization domain of H-NS. Hha molecules bind the opposite surfaces of an H-NS dimer, with minimal structural rearrangements occurring during complex formation. The results support the model in which Hha molecules cover the opposite surfaces of the H-NS protein fibers, regulating transcriptional repression and acting as an intermediary in H-NS gene suppression by the positively charged residues exposed on the surface [151].

### 3.7. The LysR Family

The last discussed family of transcriptional regulators associated with virulence of pathogenic *Salmonella* is the LysR family (LTTR) [40,156]. LysR activates transcription of regulons and operons, controlling metabolic pathways of response to oxidative stress and pathogen virulence [40]. LysR forms homotetramers and requires an inductor to activate the expression of target genes [157]. The N-terminal region contains a DNA binding domain (DBD) with an HTH motif specifically binding to the promoter region of target genes. On the other hand, the C-terminus of LysR is connected with DBD via a linker helix, forming a regulator domain (RD) containing a pocket for the binding of inductor molecules [40,158,159]. The characteristics of the LysR family with their assigned role in the virulence of *Salmonella* are presented in Table 3.

#### 3.7.1. SpvR

The SpvR factor encoded by the main virulence plasmid of *S. enterica* is a positive regulator of the LTTR system [160,162]. The *spvR* is located directly before the *spvABCD* operon and codes the DNA-binding SpvR protein. The activity of SpvR is required to express pathogen virulence, mediated by *spv* [160]. The SpvR binds to sequences above its own promoter and above the *spvA* promoter in the presence of the alternative RpoS sigma factor. The expression of the *spv* operon is induced by the intracellular environment of host cells and depends on both SpvR and RpoS activity [161]. The virulence phenotype in *locus spv* requires an increase in the *spvR* and *spvBC* expression [168]. The *spv* operon displays susceptibility to changes in DNA conformation and is suppressed by the H-NS system [162].

#### 3.7.2. LeuO

In *Salmonella*, the LeuO is an important transcription regulator of genes coding virulence factors that are suppressed by the H-NS system [162]. The LeuO acts as an H-NS antagonist by activation of appropriate genes, which are H-NS repressed [158]. Moreover, LeuO suppresses SPI-1 expression by direct activation of the *hilE* promoter and an unknown mechanism independent of the HilE protein. It has been suggested that LeuO also acts as a backup to H-NS. The repression of SPI-1 genes may occur under conditions in which the H-NS proteins do not exhibit such activity [169]. The expression of the *leuO* is induced by the ToxR, which is associated with *Salmonella* resistance to bile salts and acids. The *leuO* is positively regulated by its own promoter as well as by the wHTH (winged helix-turn-helix) protein named LrhA [162].

#### 3.7.3. LoiA

The LoiA coded by SPI-14 is involved in the low oxygen (O_2_) signaling pathway. Under low O_2_ level conditions, LoiA can positively regulate the HilA protein and SPI-1 genes by directly activating HilD transcription, increasing *S*. Typhimurium ability to invade epithelial cells [48]. Li et al. [163] reported that LoiA induced transcription of genes associated with the virulence of *S*. Typhimurium via impacting the ATP-dependent Lon protease, a negative regulator of the SPI-1 genes. It has been shown that during pathogen invasion, LoiA is responsible for suppressing the *lon* expression by binding to the *lon* promoter region [163]. In *S*. Typhimurium, induction of the SPI-1 gene expression by LoiA is correlated with its repressive activity toward *lon*, enabling invasion and adhesion of *Salmonella* to the intestinal epithelium cells [170].

#### 3.7.4. STM0030

The use of carbohydrate transport and metabolism increases intracellular survival and proliferation of *Salmonella*. During intracellular colonization, *Salmonella* may use several carbon sources, including inter alia, glycerol, fatty acids, *N*-acetylglucosamine, gluconate, glucose, lactate, and arginine. Zhang et al. [40] showed that STM0030 can influence intracellular replication and virulence by controlling the expression of the allantoin catabolic genes such as *allP* and *allB*. Furthermore, the effect of STM0030 on intracellular replication and virulence by modulating SipA expression (encoded by SPI-1) was also reported.

#### 3.7.5. CysB

The CysB regulator plays a pivotal role in the activation of the cysteine regulator (Cys) [164,171]. CysB is a positive regulator of the gene expression involved in the sulfate metabolism and cysteine biosynthesis pathways. Activation of the CysB is correlated with the limited availability of the sulfur levels by *O*-acetyl-serine (OAS) and *N*-acetylserine (NAS) in the host environment [164]. According to VanDrisse and Escalante-Semerena [171], *N*-acetyl-serine (NAS), the product of spontaneous O-N migration of the acetyl group of OAS, induces the transcription of the cysteine operon.

#### 3.7.6. LtrR

The LtrR is responsible for the induction of *ompR* expression, which is an important regulator of OmpC and OmpF associated with resistance of *Salmonella* to sodium deoxycholate [34]. The induction of the *ltrR* expression is regulated by two alternative promoters, leading to the generation of two different transcripts [166]. One of the transcripts, *ltrR,* is repressed in the promoter and the coding regions by the H-NS system, whereas Lrp inhibits its expression in the coding region. The second transcript, *ltrR*, is repressed only in the coding region by H-NS and Lrp. pH 7.5 is a positive signal involved in the expression of both *ltrR* transcripts.

#### 3.7.7. Hrg

The presence of Hrg (hydrogen peroxide resistance gene) regulon in the genome of *Salmonella* is correlated with protection against oxidative stress and survival inside macrophages [156,167]. Hrg is one of the defense lines of pathogens against the immune reaction of the host-based oxidants by disrupting the formation of reactive oxygen species (ROS) [156].

## 4. Summary

Most of the antibiotics available on the market today come from the 1980s, the golden age of antibiotic therapy. We are currently experiencing a huge disproportion between the demand for new drugs and their supply. Meanwhile, according to WHO, the post-antibiotic era has already begun. Incorrect and frequent use of broad-spectrum antibiotics, without detailed knowledge of the biological characteristics of the pathogen, results in an increase in resistance to the drugs used. The pathogen resistance to antibacterial compounds is based on a few major strategies such as inactivation of the compound, the mechanism of its active removal from the cell, modification of the site of action, and changes in the permeability of cellular sheaths. Molecular mechanisms such as gene expression, post-transcriptional modification, and protein translation are crucial pathways of pathogen multidrug resistance. In *Salmonella*, various signaling pathways positively regulate the expression of resistance genes in response to specific environmental stimuli, e.g., the presence of certain antibiotics and metal ions. Therefore, a more comprehensive investigation of the molecular mechanisms determining the acquisition of drug resistance by pathogenic *Salmonella* strains on the molecular level is thus justified. Perhaps, over time, the right strategy in medicine will be to return not to pharmacological but biological methods of combating pathogenic bacteria.

Currently, some research centers investigate the effectiveness of using preparations based on bacteriophages as an alternative to the commonly used antibiotics. There are commercial products based on lytic bacteriophages available on the market, dedicated not only to therapy in medicine but also to the food industry. Effective inhibition of molecular mechanisms of these regulatory systems could, if not replace, supplement classic antibiotic therapy in the future and contribute to effective combating infections caused by multidrug-resistant *Salmonella* strains.

## Figures and Tables

**Figure 1 pathogens-10-00801-f001:**
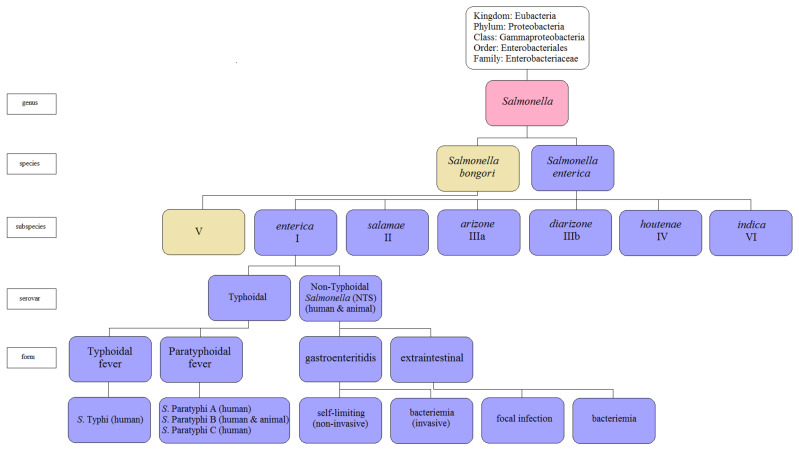
The current *Salmonella* taxonomy (adapted with permission from Hurley et al. [12]).

**Figure 2 pathogens-10-00801-f002:**
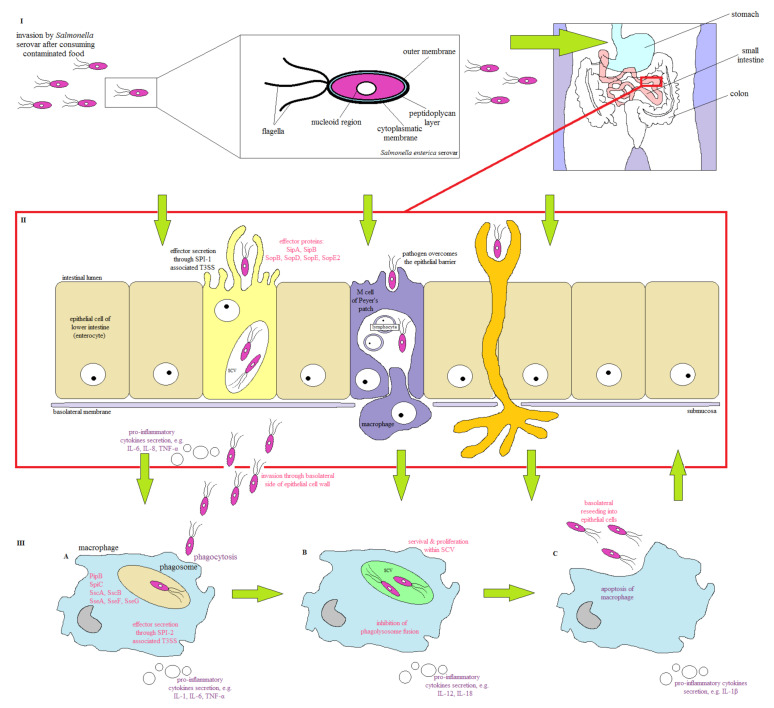
The strategy of host infection by *Salmonella* (adapted with permission from Hurley et al. [12]). (**I**) *Salmonella* infection most often occurs via the consumption of food contaminated with these pathogens. The complex structure of the *Salmonella* cell membrane and activation of ATR allows bacteria to survive in an acidic environment and overcome the barrier of the intestinal mucosa. In the colonization process, *Salmonella* uses flagella and chemotaxis to target cells. (**II**) After entering the intestine, bacteria adhere to the mucosa of the epithelial lining of the intestine and M cells by the adhesive agents present on their surfaces. Bacteria can cross the epithelial barrier by passive transport or by active invasion. One site of invasion is the apical side of the M cells with Peyer′s patches, where bacteria migrate to the underlying alveoli and kezoidal lymph nodes of the lymph tissue. The active invasion of epithelial cells (enterocytes) occurs as a result of the secretion of SPI-1-encoded effector proteins by T3SS, which reorganize the cytoskeleton and create membrane wrinkles. *Salmonella* can be taken up directly by dendritic cells from the mucosa, alternatively. (**III**) (**A**) After passing through the epithelial barrier, *Salmonella* is absorbed by the macrophages by phagocytosis. Effector proteins secreted by macrophages into the cytosol prevent phagosome fusion with the lysosome. (**III**) (**B**) Internalization by macrophages causes *Salmonella* to be localized in the SPV where the pathogen is protected from the host′s protective mechanisms. Bacteria proliferate, causing macrophages to secrete cytokines. (**III**) (**C**) The pathogens present in macrophages lead to the secretion of pro-inflammatory cytokines and the formation of an inflammatory reaction. Cytokines protect against infection by interfering with the host′s defenses. Macrophage apoptosis occurs, and the pathogen emerges from the cell re-penetrates into epithelial cells and other phagocytic cells of the host.

**Table 1 pathogens-10-00801-t001:** Functions of SPIs in the pathogenicity of *Salmonella*.

Pathogenicity Island	Virulence Factor	Function	Reference
SPI-1	SipA, SipB	invasion of intestinal epithelial cells; development of SCV; encoding the effector proteins important for actin cytoskeleton rearrangements; induction of IL-8 and *Salmonella*-elicited epithelial chemoattractant secretion; membrane ruffling that	[12,33,37,38,39,40,41,42]
SPI-2	SseF, SseG	mediates processes required for bacterial replication within host macrophages; SCV localization; inhibition of fusion between lysosomes and SCVs; avoidance of NADPH oxidase-dependent killing by macrophages; encoding of effector, chaperon and translocon proteins	[12,33,37,38,39,40,43]
SPI-3	MisL	host cell attachment and long-term colonization; intramacrophage survival	[37,38]
SPI-4	SiiE	mediation in adhesion to epithelial cells; involvement in intracellular survival and systemic infection by *S*. Typhimurium	[12,37,38]
SPI-5	SopB	important for *S*. Dublin virulence; involvement in intracellular survival and systemic infection by *S*. Typhimurium; membrane ruffling; activation of pro-inflammatory response	[12,37,38,44]
SPI-6	Tar4, SciG, SciS	combating host resistance to pathogen colonization; involvement in intracellular survival and systemic infection by *S*. Typhimurium	[37,45]
SPI-7	-	virulence factor for typhoid fever	[46]
SPI-11	PagC, PagD, EnvE, EnvF	involvement in intracellular survival and systemic infection by *S*. Typhimurium; resistance to antimicrobial peptides; survival within macrophages	[37,38]
SPI-12	SspH2	intracellular survival	[38]
SPI-13	STM3118, STM3119	survival within macrophages	[38]
SPI-14	LoiA	encoding the LoiA protein necessary for the invasion of the intestinal epithelium by *S*. Typhimurium; regulation of SPI-1 genes	[37,38,40]
SPI-16	STM0557	long-term colonization	[38]
SPI-19	-	necessary for survival within macrophages and initial colonization by *S*. Pullorum in chicken	[47]

**Table 2 pathogens-10-00801-t002:** Two-component systems targeted on antibiotic resistance in *Salmonella*.

TCS System	System Activation Signals	Components of System		Antibiotic Resistance	Function	Reference
		HK	RR			
PhoPQ	low Mg^2+^; acidic pH; cationic antimicrobial peptide (CAMP); hyperosmotic stress	PhoQ	PhoP	polymyxin	LPS modifications (by adding 4-aminoarabinose to lipid A); downregulation of the expression of genes located on SPI-1; increasing resistance to both conventional antibiotics and antimicrobial peptides	[116,117,118]
CpxAR	acidic pH	CpxA	CpxR	β-lactam	resistance to β-lactam antibiotics in the absence of AcrB efflux pump; downregulation of the MDR-related genes expression; decreasing porin expression	[119]
BaeSR	spheroplasting and exposure to indole, tannin, zinc, or cooper	BaeS	BaeR	ceftriaxone	upregulation of MDR efflux pumps; drug resistance by regulating the gene expression encoding drug transporters	[120]
EnvZ/OmpR	acidic pH	EnvZ	OmpR	β-lactam	activation of the *ssrA* transcription and production of SsrA and SsrB; decreasing porin expression, upregulation of MDR efflux pumps	[8,121]
PmrAB	acidic pH; high Fe^3+^/Al^3+^; activation by the PhoPQ system	PmrA	PmrB	polymyxin	LPS modifications (modification of lipid A)	[122]

HK—histidine kinase; RR—response regulator.

**Table 3 pathogens-10-00801-t003:** The role of proteins from the LTTR family in the virulence of *Salmonella*.

LTTR System	Function	Reference
SpvR	enhancing bacterial virulence by inhibiting autophagy; inducing the *spv* operon expression in the intracellular environment of host cells	[160,161]
LeuO	forming tetramer residues involved in oligomerization, DNA binding, transcriptional regulation	[158,162]
LoiA	encoded by SPI-14; essential for the *Salmonella* invasion of intestinal epithetical cells	[40,163]
STM0030	regulating genes essential for *Salmonella* intracellular replication and virulence	[40]
CysB	activator of the Cys regulon under the sulfur limitation	[164,165]
LtrR	indirect regulation of the porin synthesis; bacterial transformation and bile resistance	[166]
Hrg	protection against oxidative stress; bacterial growth and survival within macrophages	[167]

## Data Availability

Not applicable.

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
