# Peer review of "Transcriptional Regulation of the Multiple Resistance Mechanisms in Salmonella—A Review"

_pathogens, 2021, doi:10.3390/pathogens10070801_

Round 1

Reviewer 1 Report

    The review article made by  Wojcicki et al. is well written. This paper reviews the transcriptional regulation of multiple drug resistance mechanism as well as pivotal mechanisms required for Salmonella infection comprehensively.  I would suggest the authors could modify the title, which is not limited to antibiotics only could fit the context adequately. I can not agree with the statement "Ceftriaxone or azithromycin are used in the case of unsuccessful therapy with fluoroquinolones" (line 69-72). This suggests fluoroquinolones are the first -line antibiotics for Salmonella infection. 

Author Response

Dear Editor of Pathogens and Reviewer,

Thank you for allowing us to submit a revised draft of the manuscript “Transcriptional regulation of the multiple resistance mechanisms in Salmonella – A review” for publication in the Pathogens.

We appreciate the time and effort that you and the reviewers dedicated to providing feedback on our manuscript and we are grateful for the insightful comments on and valuable improvements to our paper. We have incorporated most of the suggestions made by the reviewers. Those changes are highlighted yellow within the manuscript. Please see below, in blue, for a point-by-point response to the reviewers’ comments and concerns. All page numbers refer to the revised manuscript file with tracked changes.

Response to Reviewer 1 Comments

The review article made by  Wojcicki et al. is well written. This paper reviews the transcriptional regulation of multiple drug resistance mechanism as well as pivotal mechanisms required for Salmonella infection comprehensively.  I would suggest the authors could modify the title, which is not limited to antibiotics only could fit the context adequately. I can not agree with the statement "Ceftriaxone or azithromycin are used in the case of unsuccessful therapy with fluoroquinolones" (line 69-72). This suggests fluoroquinolones are the first -line antibiotics for Salmonella infection. 

In line with the reviewer's suggestion, we modified the title of the manuscript in the contest of adequately in Salmonella“Transcriptional regulation of the multiple resistance mechanisms in Salmonella – A review”.

Moreover, we re-written the sentences about salmonellosis therapy using different antibiotics, page 9, line 67-72. 

Reviewer 2 Report

This review sheds light on the currently available molecular mechanisms that govern genes expression involved in multi-drug resistance Salmonella. The review can add an important data to enrich the knowledge in this area; however, minor corrections are needed. 

-The review has very poor referencing as there are several long texts with only one reference particularly the introduction. For example, lines 70-78 zero reference.

-Line 20, something wrong with a word [said]? It has to be different thing.

-Line 24, [expression genes] should be [genes expression].

-Line 30, its unclear, please re-write the sentence to delineate the importance of this review.

-Lines 35-36, you should put comma after [Salmonella] and the second comma after [family].

-Lines 41-43, please be careful as Typhoid fever caused by S. Typhi not enetrica, revise your info and re-write the sentences.

Line numbers after figure 1

-Line 4, after figure 1, add [s] to [cross].

-Line 6, replace [involves] with [uses].

-Line 7, strike off [Salmonella].

-Lines 63-76, be direct,  too much irrelevant text.

-Line 67, FQ are not membrane active antibiotics, please revise the MOD of FQ.

-Line 71, Letter I should be small in [In].

-In table 2, the PhoPQ system that regulates LPS modification is common with peptides antibiotics such as polymyxins, not common with the aminoglycosides, revise references 84-86. Same for PmrAB and check ref 90.

-Page 17, lines 67-68, re-write as the info is not correct.

Author Response

Dear Editor of Pathogens and Reviewer,

Thank you for allowing us to submit a revised draft of the manuscript “Transcriptional regulation of the multiple resistance mechanisms in Salmonella – A review” for publication in the Pathogens. We appreciate the time and effort that you and the reviewers dedicated to providing feedback on our manuscript and we are grateful for the insightful comments on and valuable improvements to our paper. We have incorporated most of the suggestions made by the reviewers. Those changes are highlighted yellow within the manuscript. Please see below, in blue, for a point-by-point response to the reviewers’ comments and concerns. All page numbers refer to the revised manuscript file with tracked changes.

Response to Reviewer 2 Comments

This review sheds light on the currently available molecular mechanisms that govern genes expression involved in multi-drug resistance Salmonella. The review can add an important data to enrich the knowledge in this area; however, minor corrections are needed. The review has very poor referencing as there are several long texts with only one reference particularly the introduction. For example, lines 70-78 zero reference.

In line with the reviewer's suggestion, additional references have been added in the Introduction and within the manuscript.

Line 20, something wrong with a word [said]? It has to be different thing.

It has been corrected.

Line 24, [expression genes] should be [genes expression].

It has been corrected.

Line 30, its unclear, please re-write the sentence to delineate the importance of this review.

A sentence has been re-written.

Lines 35-36, you should put comma after [Salmonella] and the second comma after [family].

It has been corrected.

Lines 41-43, please be careful as Typhoid fever caused by S. Typhi not enetrica, revise your info and re-write the sentences.

Sentences have been re-written.

Line numbers after figure 1

Line 4, after figure 1, add [s] to [cross].

It has been corrected.

Line 6, replace [involves] with [uses].

It has been corrected.

Line 7, strike off [Salmonella].

It has been corrected.

Lines 63-76, be direct,  too much irrelevant text.

In line with the reviewer's suggestion, the text has been shortened to the most important information.

 Line 67, FQ are not membrane active antibiotics, please revise the MOD of FQ.

The mistake has been corrected.

Line 71, Letter I should be small in [In].

It has been corrected.

-In table 2, the PhoPQ system that regulates LPS modification is common with peptides antibiotics such as polymyxins, not common with the aminoglycosides, revise references 84-86. Same for PmrAB and check ref 90.

The mistake has been corrected and revision of the references was done.

Page 17, lines 67-68, re-write as the info is not correct.

It has been corrected.
